# Biohydrometallurgy for Rare Earth Elements Recovery from Industrial Wastes

**DOI:** 10.3390/molecules26206200

**Published:** 2021-10-14

**Authors:** Laura Castro, María Luisa Blázquez, Felisa González, Jesús Ángel Muñoz

**Affiliations:** 1Department of Applied Mathematics, Materials Science and Engineering and Electronic Technology, School of Experimental Sciences and Technology, Rey Juan Carlos University, 28935 Móstoles, Spain; 2Department of Chemical and Materials Engineering, University Complutense of Madrid, 28040 Madrid, Spain; mlblazquez@quim.ucm.es (M.L.B.); fgonzalezg@quim.ucm.es (F.G.); jamunoz@ucm.es (J.Á.M.)

**Keywords:** bioleaching, rare earth elements, recycling, wastes

## Abstract

Biohydrometallurgy recovers metals through microbially mediated processes and has been traditionally applied for the extraction of base metals from low-grade sulfidic ores. New investigations explore its potential for other types of critical resources, such as rare earth elements. In recent times, the interest in rare earth elements (REEs) is growing due to of their applications in novel technologies and green economy. The use of biohydrometallurgy for extracting resources from waste streams is also gaining attention to support innovative mining and promote a circular economy. The increase in wastes containing REEs turns them into a valuable alternative source. Most REE ores and industrial residues do not contain sulfides, and bioleaching processes use autotrophic or heterotrophic microorganisms to generate acids that dissolve the metals. This review gathers information towards the recycling of REE-bearing wastes (fluorescent lamp powder, spent cracking catalysts, e-wastes, etc.) using a more sustainable and environmentally friendly technology that reduces the impact on the environment.

## 1. Introduction

Rare earth elements (REEs) are strategic metals that facilitate the transition from the current economy based on fossil fuel to an efficient circular economy based on renewable energy. These metals are often needed in small quantities; however, they are essential for fabricating a large amount of technologically smart products for electronic, optical, and magnetic applications [1].

Most of rare earths are common elements in the Earth’s crust, and some of them are even more abundant than other metals, such as copper or lead. Despite their moderate abundance, rare earth elements are scarcely concentrated in mineral deposits, hampering their extractive metallurgy, which is complex and demands economic solutions. The production of REEs is growing exponentially since their discovery in the 18th century, with a significant rise over time from 1000 t in 1930 to 133,600 t in 2010 [2]. The increasing demand for REEs has led an escalating production as well.

REE resources are mostly present in oxidic form, mainly as rare earth oxides, phosphates, carbonates, and silicates, due to their strong affinity for oxygen. Recent estimates indicate that 100 Mt of rare earth oxides are accessible in more than thirty countries all over the world. More than 200 REE-bearing mineral ores have been identified; nevertheless, only three of them are considered mineral ores for economic extraction: bastnasite ((Ce,La)(CO_3_)F), monazite ((Ce,La,Nd,Th)PO_4_), and xenotime (YPO_4_) [3]. Consequently, the primary sources are disseminated worldwide but are confined mainly in China, Australia, and USA.

Moreover, REEs are also present in industrial wastes in great amounts, and industrial wastes have been considered as a potential resource for these critical metals [4,5,6]. Phosphogypsum is one of the most remarkable REE-bearing wastes generated during the wet phosphoric acid process of fertilizer production. Red mud residues from the digestion of bauxites in the Bayer process are also rich in valuable rare earth metals and their recovery can be economically valuable.

Additionally, some post-consumer wastes can be recycled due to their significant quantities of REE, such as magnets (38%), lamp phosphors (32%), and metal alloys (13%). These materials comprise more than 80% of the REE market. Modern fluorescent lamps typically retain more than 20% (*w*/*w*) REE (Ce, Eu, La, Tb, and Y) [7].

After ore and/or industrial waste concentration processing, rare earth metals are dissolved selectively from raw materials using acid (H_2_SO_4_, HCl, HNO_3_, H_3_PO_4_) or alkaline (Na_2_CO_3_, NaHCO_3_) reagents under high temperatures, and this could pose an environmental problem. Actinides, such as uranium and thorium, with similar chemical properties to REEs, are often co-dissolved during hydrometallurgical processes, leading to a complex methodology [8,9,10].

Biohydrometallurgy has been successfully applied at the industrial level for the recovery of metals such as uranium, copper, and gold [11,12]. Biohydrometallurgical processes are usually applied to materials that would not be feasible to mine or treat using conventional chemical methods and would be considered residues. Consequently, these technologies could play a fundamental role in the treatment of REE-bearing wastes since they offer an alternative to physicochemical-based methods. Furthermore, the bioleaching of REEs would be involved in the development of more cost-effective, less energy demanding, and less polluting metal extraction processes than pyro- and hydrometallurgical processes. These biotechnological processes take place through interactions between microorganisms and metal-bearing ores that dissolve valuable metals. REE recovery from solid resources has been investigated with a wide range of microorganisms, both autotrophic and heterotrophic, and using both pure and mixed microbial cultures [13,14,15].

This review provides an insight into the global situation of REEs and the potential application of microorganisms in the extraction of REEs from industrial residues.

## 2. World Market and Relevance of REEs

There is a concern among many countries about REEs and REE supply chains because these metals are used in small quantities for a variety of economically significant applications, such as laptops, cell phones, electric vehicles, renewable energy capture technologies, and other industrial products. In 2018, the U.S. Department of the Interior and other executive branch agencies released a list of 35 critical minerals, including rare earth elements [16]. Likewise, the European Commission developed a critical assessment of non-energy and non-agricultural raw materials in 2017 that included heavy rare earth elements, light rare earth elements, and platinum group metals [17].

World mine production was estimated to be 240,000 tons of rare earth oxide (REO) equivalent in 2020, which means an increase of 11% compared with 2019 (Figure 1a). China dominates the global supply of rare earth minerals, separated compounds, and metals. China exports REEs to the United States (31%), Japan (27%), the Republic of Korea (11%), the Netherlands (9%), and Germany (6%). Other countries are making efforts to develop domestic supplies of critical materials and to encourage the domestic private sector to produce and process these materials. For example, United States enhanced its production, all of which was exported, by 36% in 2020 compared with 2019 [18]. Nevertheless, some raw materials do not exist in economic quantities, and so processing and manufacturing may not be cost competitive.

Despite their relative abundance in the Earth’s crust, REE resources with minable concentrations are less common. At the present time, about 850 minable REE deposits have been identified in a few locations, which are mainly in China, Vietnam, Brazil, Russia, India, and Australia (Figure 1b) [18,19].

The falling trend in the price of rare earth products that began after prices spiked in 2011 was reversed. Prices for most REEs are rising compared with those in 2016 (Figure 2). The price of gadolinium, praseodymium, and neodymium experienced the greatest increase, whereas the yttrium and dysprosium prices declined. The estimated unit value of rare-earth compounds was USD 11.60 per kg in 2017, based on information from the U.S. Census Bureau on imports [20].

The global calculated consumption of rare earth varies significantly owing to the limited data transparency, and it generally ranges between 140,000 and 170,000 tons of REO equivalent. In addition, the global consumption of scandium was estimated to be 10–20 tons per year [21]. The quantity of specific REEs used has a strong link to the market sector and application. For instance, lanthanum and cerium, and lower amounts of neodymium, are used in the catalysts sector. There are several types of permanent magnets, but neodymium–iron–boron magnets are fabricated with neodymium and praseodymium, and samarium–cobalt magnets consume samarium and gadolinium. Batteries contain mainly lanthanum, and ceramics contain yttrium. Europium, yttrium, and terbium are usually associated with the phosphors sector.

World growth rate of REE consumption is foreseen to be 5–7% per year through 2022. The magnet materials sector is expected to grow more than other markets, such as catalysts, ceramics, or phosphors. The rising demand for REEs as well as the implementation of environmental and production regulation beyond 2022 led to an increase in REE prices, and this situation may therefore drive the mining and processing development outside of China.

## 3. Biorecovery of REEs from Industrial and Electronic Waste

The recycling of precious and base metals has achieved high rates; nevertheless, the recycling rates of REEs are still very low (<1%). This fact can be explained by different elements, such as technological issues, low environmental impact of the REEs, and, until recently, low prices and lack of incentives. The technological difficulties of the recycling of rare earths are caused by the low concentrations of these elements in consumer products [22].

The growing generation of industrial and electronic residues and their remarkable content in critical metals has led to these materials becoming an alternative economic source of REEs. Recently, secondary sources of REEs, including industrial wastes, mine wastes, and electronic wastes, are being treated using bioprocess technology for metal recovery. Most of the REE bioleaching studies that have been developed have treated mineral ores with a variety of microorganisms. Phosphate-solubilizing microorganisms are particularly relevant in the bioleaching of monazite because they transform insoluble phosphate into more soluble phosphorous forms [3,23,24]. The mineral phosphate solubilization occurs through the biological generation of organic acids and the proton substitution reaction:[M^n+^][PO_4_^3−^] + [HA] = [H^+^] [PO_4_^3−^] + [M^n+^] [A^−^](1)
where HA is the organic acid produced by the microorganism, and M^n+^ is a metallic cation. The mineral phosphate solubility, the number of protons in the organic acid, and its pKa influence the efficiency of the solubilization.

Bioleaching studies of REE extraction from wastes are still in their infancy. These residues have become a valuable alternative for REE recovery due to mineral scarcity and environmental degradation, and the development of bioprocess technology has a crucial role in sustainable mining for the green economy (Table 1).

### 3.1. Phosphogypsum

Phosphogypsum is an industrial residue of relevance for REE recovery. REEs are frequently associated with phosphate deposits, and the wet phosphoric acid process from the fertilizer industry generates phosphogypsum waste in vast amounts (100–280 Mt per year). Estimates suggest that 21 Mt of REEs are enclosed in the total of phosphogypsum waste accumulated to date [2].

The bacterium *Gluconobacter oxydans* grown on glucose generates a biolixiviant containing organic acids that have been used for the leaching of synthetic phosphogypsum doped with six rare earth elements (yttrium, cerium, neodymium, samarium, europium, and ytterbium). The lixiviant produced by the bacterial culture mainly contains gluconic acid (220 mM) and the pH value is 2.1. REEs leaching yield from phosphogypsum using the biolixiviant were compared with the results obtained using sulfuric acid, phosphoric acid, and commercial gluconic acid. The bioreagent produced by *Gluconobacter oxydans* was more efficient for REE dissolution than commercial gluconic acid and phosphoric acid but less efficient than sulfuric acid [25].

Salo et al. [26] studied the combination of REE dissolution from phosphogypsum waste using sulfuric acid with the leachate treatment in a sulfate-reducing bioreactor to recover REEs and remove sulfate from the residual solution. During the process, sulfate could be transformed to sulfide and acidity neutralized by the production of carbonate:2 CH_3_CHOHCOOH + 3SO_4_^2−^ → 3H_2_S + 6HCO_3_^−^(2)

Moderate REE leaching began at 0.01 M H_2_SO_4_ concentration, reaching a yield between 6% and 15%, and increased continuously when increasing the H_2_SO_4_ concentration to 0.05 M (yield 34–62%). However, the bioreactor requires rather a mild leachate (<0.02 M H_2_SO_4_) while ensuring the growth of the microorganisms and the precipitation of REEs. Finally, the precipitate from the bioreactor was collected, evidencing an accumulation of REEs, and the separation of recoverable REE phases seems to be possible (Figure 3). 

### 3.2. Red Mud

Red mud is waste material from the digestion of bauxites in the Bayer process that could be considered as a secondary REE resource. It is estimated that around 2700 Mt of red mud residues have been accumulated in Bayer plants around the world, and its production increases at a rate of 120 Mt per year [37]. These residues are hazardous due to their alkalinity, but they could also be economically treated for their richness in valuable rare earth metals. The main element in red mud is scandium (95% of the economic value of the REE), with a content between 130 and 390 ppm [38].

A filamentous acid-producing fungus isolated from bauxite residues, *Penicillium tricolor* RM-10, was used for the REE-bioleaching. The highest extraction yields were achieved under a two-step process at 10% (*w*/*v*) pulp density due to the production of citric and oxalic acids [27]. Other work developed with the chemoheterotrophic bacterium *Acetobacter* sp. recovered 53% of Lu, 61% of Y, and 52% of Sc in a one-step process at 2% pulp density. Furthermore, this bacterium can be used at high pulp densities of red mud because the secretion of organic acids by the microorganisms increased with the waste concentration [28].

Recently, Kiskira et al. [29] investigated the bioleaching of red mud using different microbial cultures and solid-to-liquid ratios. The maximum extraction yield of Sc was 42% using *Acetobacter tropicalis* in a one-step bioleaching process at 1% pulp density. The bioleaching results suggest a synergistic effect of the different organic acids (acetic, oxalic, and citric) produced by microorganisms.

### 3.3. Cracking Catalysts

Oil refining and biocombustible industries generate around 700,000–900,000 tons of spent fluid catalytic cracking catalyst per year worldwide, and the management of these solid wastes and the recycling of rare earth metals have become a challenge [39]. The major REE present in cracking catalysts are cerium and especially lanthanum. The use of cell-free culture supernatants of *Gluconobacter oxydans* containing gluconic acid for REE dissolution from spent cracking catalysts allowed the recovery of 49% of the total REE (preferentially lanthanum over cerium) [4]. Further studies optimized this biolixiviant reaching a yield up to 56% of REE and a continuous bioreactor system was developed to achieve efficiencies up to 51% [30]. Finally, techno-economic analysis showed that the use of agricultural wastes as substrate for bacterial growth instead of glucose provides a cost-effective process for REE recovery [31].

The fungus *Aspergillus niger* was also investigated for leaching spent cracking catalysts at 1, 3, and 5% pulp densities obtaining 63%, 52%, and 33% of lanthanum recovery, respectively. The efficiency of the use of the cell-free supernatant as a lixiviant reagent at 1% pulp density resulted in 30.8% of leaching recovery [32].

A biochemical process using *Yarrowia lipolytica* IM-UFRJ 50,678 to recover REEs from spent catalysts employing biodiesel-derived glycerin as the main carbon source reached yields of 53% of La and 99% of Ce and Nd, at 50 °C [33].

Recent advances reported the bioleaching of REEs from spent catalytic cracking catalyst in the presence of Fe(II) using *Acidothiobacillus ferrooxidans*, an iron-oxidizing bacterium [34]. Higher recoveries of La were obtained at 1% pulp density, and higher leaching efficiencies for Ce bioleaching were reached at 5% and 7% pulp densities. The maximum recovery of La and Ce was 83% and 23%, respectively. This research recommended bioleaching for La in comparison with chemical leaching.

### 3.4. Fluorescent Lamps

Compact fluorescent lamps are end-of-life products worth mentioning in relation to REE recovery. Their composition on average is glass (88 wt.%), metals (5 wt.%), plastic (4 wt.%), lamp phosphor powder (3 wt.%), and mercury (0.005 wt.%). The lamp phosphor powder contains approximately 10% of rare-earth phosphors bound in the triband dyes [40]. Hence, the removal of lamp phosphors waste containing REEs with mercury would lead not only to a loss of resources but also to environmental risks. Some countries collect great amounts of fluorescent phosphor powder as a separated fraction from the recycling of fluorescent lamps (175 tons per year in Germany), becoming a secondary resource of REE.

The most common rare-earth phosphors in lamp powder are Y_2_O_3_:Eu^3+^, LaPO_4_:Ce^3+^, (Gd,Mg)B_5_O_12_:Ce^3+^, Tb^3+^, (Ce,Tb)MgAl_11_O_19_, and BaMgAl_10_O_17_:Eu^3+^ [40], and there is a growing interest in the extraction of REEs from these compounds through microbial processes (Figure 4).

Hopfe et al. [35] investigated a symbiotic mixed culture from tea Kombucha, consisting of yeasts and acetic acid bacteria, for dissolving REEs from fluorescent lamp waste. The microbial species *Zygosaccharomyces lentus* and *Komagataeibacter hansenii* were isolated; however, the highest extraction yields were achieved using the entire Kombucha-consortium or its supernatant as lixiviant, compared with experiments using the isolates. The pH lowered during the microbial growth because of the production of acetic and gluconic organic acids.

Further studies reported the evaluation of a broad spectrum of different microorganisms for their potential to leach REEs from lamp residues. The higher leaching yields were obtained with the strains *Komagataeibacter xylinus*, *Lactobacillus casei*, and *Yarrowia lipolytica*, achieving a total release of REEs of 12.6%, 10.6%, and 6.1%, respectively. Yttrium and europium were selectively dissolved during the experiments [36].

Moreover, the supernatant produced by the bacterium *Gluconobacter oxydans* was also investigated as a biolixiviant for the recovery of REEs from phosphor powder; nevertheless, only the 2% of the total REEs was leached [4].

The biosolubilization of phosphor powder residues is likely associated with the carboxyl-functionality or a proton excess generated by the organic acids. Among the different REE components, the red dye Y_2_O_3_:Eu^3+^ was shown to be preferentially mobilized in accordance with the higher solubility of REE-oxides compared with REE-phosphates and aluminates.

### 3.5. Electronic Waste

Electronic wastes come from scrapped devices that are at the end of their economic use and consumers cannot utilize them anymore. The total generation of e-waste in 2021 is expected to reach 52.2 Mt worldwide. The biggest economic interest is centered on gold, with 50% of the possible revenue, but e-wastes contain other critical metals, such silver, copper and the rare earth elements lithium and cobalt.

There are a growing number of projects and publications about biotechnical metal extraction from e-waste. Research on bioleaching associated with printed circuit boards (PCB) recycling has mainly centered on copper and gold recovery. Cyanogenic bacteria or fungi have been investigated to recover gold from e-wastes [41]. Nevertheless, an up-scaling and optimization of these processes are required. Ferric iron produced by iron-oxidizing bacteria can also contribute to copper dissolution. Recently, a two-step reactor was designed to separate the production of biogenic ferric iron from the valuable metals, and the leaching reaction achieving a 96% extraction of Cu [42]. Many post-consumer devices contain magnets with a significant amount of REE, 20–30%. The content of Nd, Dy, and Pr in NdFeB magnets is 259.5, 42.1, and 3.4 ppm, respectively [43]. *Acidithiobacillus ferrooxidans* and *Leptospirillum ferrooxidans* have demonstrated high leaching efficiencies when grown not only in the presence of magnets but also in abiotic controls. Thus, leaching could mainly take place through chemical processes in the presence of H_2_SO_4_. Furthermore, experiments with an iron addition reached higher leaching yields because of the catalytic effect of ferric ions [44]. Moreover, biodismantling is a new application of bioleaching in the recycling of e-waste to enhance the concentration of critical and precious materials contained in the electronic components. When achieving sufficient classification of the components after separation, some rare earth elements may become economically viable because their grade would be similar to commercial ores. A concentration of 9000 µg/g of dysprosium was detected in one of these separated fractions [45].

The objective of many bioprocesses designed for the recovery of metals from wastes is the use of wastes as inputs to close a loop and diminish costs. The competitiveness and economic viability of biological extraction of metals from residues depend on different factors, such as more restrictive environmental regulations and incentives for resource recovery from waste.

## 4. Conclusions

The demand for REEs is growing due to their unique properties, and REE extraction is becoming an important issue. Biohydrometallurgy could contribute by alleviating challenges related to the scarcity of economic ore resources and an almost monopolistic market. Novel studies on biohydrometallurgy offer the possibility of facilitating the extraction of REEs from waste, increasing the number of commodities of critical materials. Bioleaching for the recovery of rare earth metals from industrial wastes can be carried out by autotrophic and heterotrophic microorganisms. Several mechanisms are involved in the mobilization of REEs: organic acids, enzymes, bacterial attachment, siderophores, etc. Moreover, the development of biotechnological strategies for the treatment of solid wastes might contribute to a sustainable economy, maximizing the number of resources and minimizing the harmful impact on the environment. Bioleaching of REEs is in its infancy, but the development of a global market and environmental policies, as well as the appearance of new drivers such as synthetic biology and digital revolution, could influence the evolution of biohydrometallurgy.

## Figures and Tables

**Figure 1 molecules-26-06200-f001:**
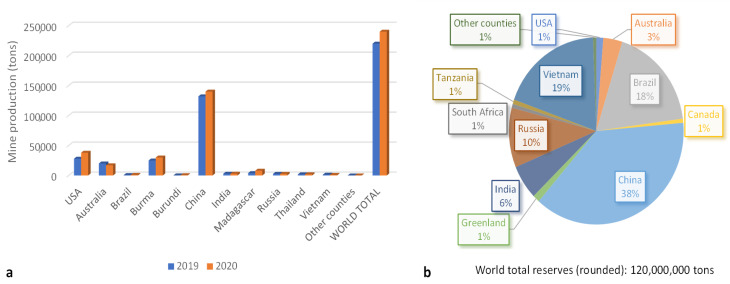
World mine production (**a**) and reserves (**b**) [18].

**Figure 2 molecules-26-06200-f002:**
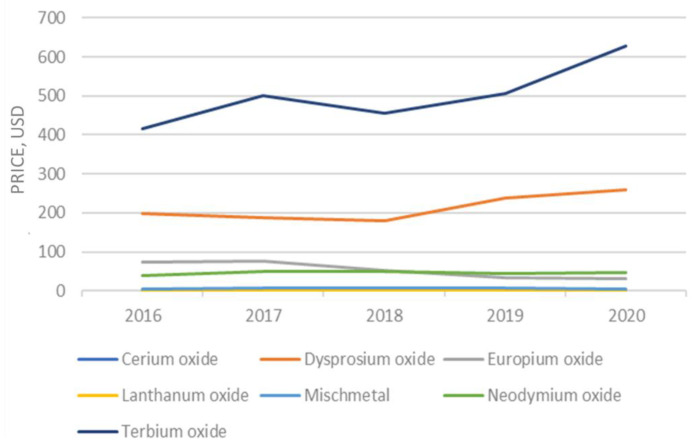
Evolution of the rare earth prices.

**Figure 3 molecules-26-06200-f003:**
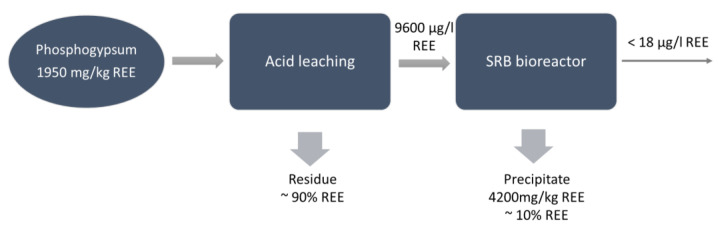
Flow chart for integrated acid leaching and biological sulfate reduction in phosphogypsum for REE recovery [26].

**Figure 4 molecules-26-06200-f004:**
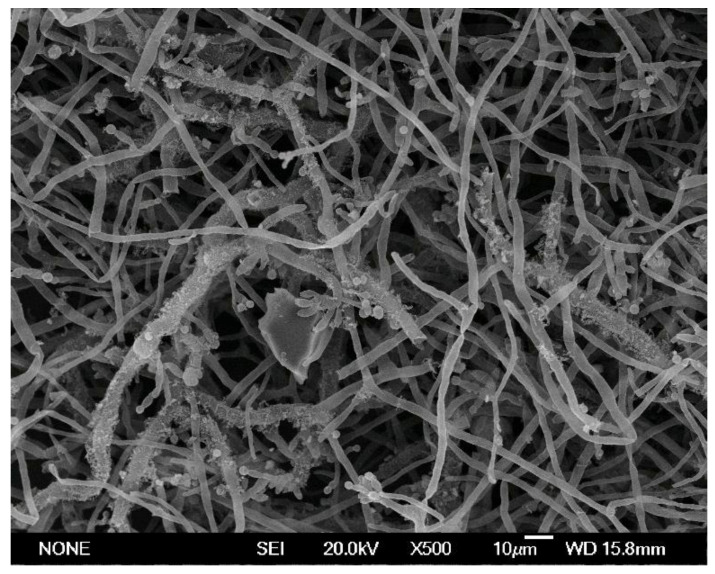
SEM image of the fungus *Aspergillus niger* grown on fluorescent lamp powder (1% pulp density) during bioleaching experiments.

**Table 1 molecules-26-06200-t001:** Summary table of bioleaching studies of REE extraction from wastes.

Type of Waste	Microorganism	REE	Reference
Phosphogypsum	*Gluconobacter oxydans* (spent medium)	Y (91.2%), Ce (36.7%), Nd (42.8%), Sm (73.2%), Eu (50%), Yb (83.75)	[25]
	Sulfate-reducing bacteria	REE concentrated phases (2.54%)	[26]
Red mud	*Penicillium tricolor* RM-10	Mainly Sc (~57%)	[27]
	*Acetobacter* sp.	Lu (53%), Y (61%), Sc (52%)	[28]
	*Acetobacter tropicalis*	Sc (42%)	[29]
Cracking catalysts	*Gluconobacter oxydans*	La, Ce (up to 56% REE)	[30,31]
	*Aspergillus niger*	La (up to 63%)	[32]
	*Yarrowia lipolytica* IM-UFRJ 50678	La (53%), Ce (99%), Nd (99%)	[33]
	*Acidothiobacillus ferrooxidans*	La (83%), Ce (23%)	[34]
Fluorescent lamps	Kombucha-consortium	7.9% REE	[35]
	*Komagataeibacter xylinus*, *Lactobacillus casei*, and *Yarrowia lipolytica*	Y, Eu (up to 12.6% REE)	[36]
	*Gluconobacter oxydans* (spent medium)	2% REE	[4]

## Data Availability

Data sharing not applicable.

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
