# Peer review of "Biohydrometallurgy for Rare Earth Elements Recovery from Industrial Wastes"

_molecules, 2021, doi:10.3390/molecules26206200_

Round 1
Reviewer 1 Report
This review summarizes the recent biotechnology related to REE recovery especially from industrial wastes. It is a very interesting review and shows that the authors are quite familiar with this topic. My suggestions to the authors are:
- Use tables to summarize the information from different studies for easier reading. For instance, the type of waste, microorganism, REE, leaching efficiency and reference could all be included.
- Use some general chemical equations to demonstrate the different mechanisms of different types of bioleaching. For instance, from ref [24], there are equations for the leaching via organic acids produced by microorganism.
- Include at least one flow chart to show the total process of bioleaching in real industrial application. Select one from any reference as a representative would be adequate.
- The content regarding Figure 2 could be expanded. Also, it would be helpful to indicate where the fungus is located on the SEM image.
Author Response
This review summarizes the recent biotechnology related to REE recovery especially from industrial wastes. It is a very interesting review and shows that the authors are quite familiar with this topic. My suggestions to the authors are:
- Use tables to summarize the information from different studies for easier reading. For instance, the type of waste, microorganism, REE, leaching efficiency and reference could all be included.
Author’s response:
The authors thank the useful comments to improve the manuscript. The table was included in the text.
- Use some general chemical equations to demonstrate the different mechanisms of different types of bioleaching. For instance, from ref [24], there are equations for the leaching via organic acids produced by microorganism.
Authors response:
Some equations have been included. Nevertheless, the available information is limited.
- Include at least one flow chart to show the total process of bioleaching in real industrial application. Select one from any reference as a representative would be adequate.
Authors’ response:
A flow chart has been included following the reviewer’s suggestion.
- The content regarding Figure 2 could be expanded. Also, it would be helpful to indicate where the fungus is located on the SEM image.
Authors’ response:
Figure 2 is a SEM image obtained by the authors. This research about REE recovery from fluorescent lamps is not published yet. The detail of the pulp density has been included. The fungal mycelium appears all over the image, it is not located at a particular point.
Reviewer 2 Report
The paper is very nice and well written and very complete, and reports on an important subject. However, it needed some minor changes.
Do not use abbreviations in the title. REE should be replaced by its meaning: rare earth elements.
Also the title should not simply mention "biotechnology" but mention specifically "biohydrometallurgy", which is the main subject of the paper.
Although the paper is well written, it lacks illustrations as only 2 figures are shown. For example, some chart with the REE prices or occurence or consumption or other important information.
Author Response
Do not use abbreviations in the title. REE should be replaced by its meaning: rare earth elements.
Authors’ response:
REE is now replaced by rare earth elements.
Also the title should not simply mention "biotechnology" but mention specifically "biohydrometallurgy", which is the main subject of the paper.
Authors’ response:
Biotechnology is now replaced by the term “biohydrometallurgy”.
Although the paper is well written, it lacks illustrations as only 2 figures are shown. For example, some chart with the REE prices or occurrence or consumption or other important information.
Authors’ response:
Thank you for your comment. New figures have been added following the reviewer’s suggestion.
Reviewer 3 Report
The manuscript I reviewed is ready for publication.
A minor point: "Extractive Metallurgy" in the text should not be capitalized
Author Response
A minor point: "Extractive Metallurgy" in the text should not be capitalized.
Authors’ response:
Thank you for the comment. The authors have made the change.